# Individual Radiosensitivity as a Risk Factor for the Radiation-Induced Acute Radiodermatitis

**DOI:** 10.3390/life12010020

**Published:** 2021-12-23

**Authors:** Juras Kišonas, Jonas Venius, Olga Sevriukova, Mindaugas Grybauskas, Daiva Dabkevičienė, Arvydas Burneckis, Ričardas Rotomskis

**Affiliations:** 1Department of Radiation Oncology, National Cancer Institute, LT-08660 Vilnius, Lithuania; mindaugas.grybauskas@nvi.lt (M.G.); arvydas.burneckis@nvi.lt (A.B.); 2Department of Neurobiology and Biophysics, Vilnius University, LT-10257 Vilnius, Lithuania; 3Medical Physics Department, National Cancer Institute, LT-08660 Vilnius, Lithuania; jonas.venius@nvi.lt; 4Biomedical Physics Laboratory, National Cancer Institute, LT-08660 Vilnius, Lithuania; ricardas.rotomskis@nvi.lt; 5Radiation Protection Centre, LT-08352 Vilnius, Lithuania; olga.sevriukova@rsc.lt; 6Biobank, National Cancer Institute, LT-08660 Vilnius, Lithuania; daiva.dabkeviciene@nvi.lt

**Keywords:** individual radiosensitivity, breast cancer radiotherapy, acute radiation dermatitis, risk factors, radiation induced skin toxicity

## Abstract

Background: Up to 95% of irradiated patients suffer from ionizing radiation (IR) induced early skin reaction, acute radiation dermatitis (ARD). Some experts think that additional skin hydration can reduce acute skin reactions. Individual radiosensitivity (IRS) determined from lymphocytes may help to predict acute radiation toxicity. The purpose of this study is to evaluate the clinical manifestation of ARD in different skincare groups during whole breast radiotherapy depending on IRS and other risk factors. Methods: A total of 108 early-stage breast cancer patients were randomized into best supportive care (BSC) and additional skincare (ASC) groups. IRS was evaluated using a G2 assay modified with caffeine-induced G2 checkpoint arrest. All patients received a 50 Gy dose to the breast planning target volume (PTV). Clinical assessment of ARD symptoms according to the CTCAE grading scale was performed once a week. Results: IRS was successfully determined for 91 out of 108 patients. A total of 10 patients (11%) had normal IRS, 47 patients (52%) were categorized as radiosensitive, and 34 (37%) as highly radiosensitive. There was no significant difference in the manifestation of ARD between patient groups by skincare or IRS. According to logistic regression, patients with bigger breasts were prone to more severe ARD (*p* = 0.002). Conclusions: The additional skincare did not improve skin condition during RT. A total of 89% of patients had increased radiosensitivity. IRS determined before RT did not show the predictive value for the manifestation of ARD. Logistic regression revealed that breast volume was the most significant risk factor for the manifestation of ARD.

## 1. Introduction

Breast cancer is the most common type of oncological disease among women, with more than 2 million new cases worldwide annually, and the most common cause of death among women with more than 600,000 deaths per year [1].

Whole breast radiotherapy (WBRT) after breast-conserving surgery (BCS) is the standard treatment for breast cancer patients due to its beneficial effect on survival, low risk of local recurrence, and acceptable cosmetic results [2,3].

Almost all irradiated patients suffer from ionizing radiation (IR)-induced early skin reaction, acute radiation dermatitis (ARD). The clinical manifestation of ARD varies from mild erythema to moist desquamation and depends on various risk factors.

Technical risk factors such as total IR dose, fractionation, etc., can be easily modified, while patient-related factors cannot. Breast size and body mass index were identified as one of the first patient dependent risk factors for ARD [4]. Later, the association with age, sex, physical activity, disease stage, smoking, etc., was noticed [5].

Researchers started to investigate how individual radiosensitivity (IRS) determined from lymphocytes may help to predict acute and late effects of radiation toxicity 20 years ago [6]. Some authors found that IRS can be a good predictor factor for acute effects of RT [7,8,9], others still have doubts [10], and this question remains relevant.

Until now, there has been a lack of clinical trials investigating the effect of topical agents for the prevention and treatment of ARD. Only a few recommendations such as wearing loose clothes, washing the skin with mild soap and water, and topical corticosteroids are universally accepted for the management of ARD [11,12,13]. Some experts think that additional skin hydration can reduce acute skin reactions [14,15].

In this article, we present results from a prospective, randomized, open-label single-center biomedical study focusing on the influence of IRS and other risk factors on ARD manifestation in breast cancer patients with different skincare during RT.

## 2. Materials and Methods

### 2.1. Patients

In the National Cancer Institute of Lithuania 110 early-stage breast cancer patients after BCS treated with whole breast irradiation were involved in this study from April 2017 to February 2020. A total of 108 patients finished the treatment.

Inclusion criteria were: age > 18 years, the multidisciplinary team decision to treat patient with RT, and performance status according to ECOG from 0 to 2. Patients with poor performance status (ECOG ≥ 3), and previously treated with RT or chemotherapy were excluded.

### 2.2. Study Methodology

The local bioethics committee approved (permission No.: 158200-17-908-418) this prospective, randomized, open-label single-center biomedical study (protocol No.: II-2016-4) in April 2017.

### 2.3. Randomization

Before RT, patients were randomized according to skincare into two groups: the best supportive care (BSC) group, and the additional skincare (ASC) group. Best supportive skincare including loose clothes and an everyday shower was prescribed for both groups from the beginning of RT For the patients in the BSC group, calendula ointment was prescribed only after the second degree of ARD was diagnosed. If ARD never reached the second degree, the calendula ointment was prescribed after RT. Patients in the ASC group were using moisturizing cream twice a day with the best supportive skincare from the beginning of RT.

### 2.4. Radiotherapy

After the diagnosis of early-stage breast cancer and surgical treatment (BCS) the whole breast RT was prescribed to all the patients following the breast cancer treatment guidelines [16]. Patients were immobilized in a supine position with hands above the head, and chest computed tomography (CT) scans were performed. Treatment planning system Eclipse^®^ version 15.5 was used for RT planning. Clinical target volume (CTV) and organs at risk (OAR) were delineated according to RTOG guidelines [17].

CT scans for RT planning were used to calculate breast volume. According to CT images, the structure of whole breast tissue was delineated for each patient. Then the volume of this structure was measured.

The RT dose was prescribed according to institutional standards and breast cancer treatment guidelines [16]. To ensure the homogeneity of the study groups, only patients that were prescribed with a total dose of 50 Grays (Gy) to the breast planning target volume (PTV) and were treated with the 3D conformal RT technique. Irradiation was delivered in daily fractions (fx) of 2 Gy per fx, 5 fx per week. Patients with other fractionation regimes were not included in this study due to differences in RT period, dose per fx and manifestation of ARD. 

### 2.5. Assessment of Skin Lesions

The CTCAE grading scale (see Appendix A) was used for clinical assessment of ARD symptoms before RT and once a week (every 5 fx). Radiation induced skin reactions were evaluated separately by the three radiation oncologists. The final degree of ARD was determined after the combination of the evaluations and when at least two evaluators agreed. Moist desquamation was evaluated once a week during the RD and up to 4 weeks after. The median follow-up was 30 days.

### 2.6. Radiosensitivity Test

The assessment of individual radiosensitivity (IRS) to IR was performed employing a modified G2 assay when the radiosensitivity of chromosomes are tested by G2 checkpoint arrest under the conventional and caffeine-containing environment. The peripheral blood was collected into the Li-heparin vacutainers before RT. The setting up of lymphocytes cultures, consisted of the addition of heparinized whole blood to enriched F-10 medium. Lymphocytes were cultured in the incubator for 72 h in a humidified air atmosphere of 37 °C in 5% CO_2_ [10]. In vitro irradiation of lymphocytes cultures to 1 Gy was followed by the end of the incubation period (RT unit T-105 X-ray; 2.3 Gy/min dose rate 23 ± 2 °C). After the irradiation, half the lymphocytes culture was supplemented with caffeine solution (4 mM) and incubated together with caffeine-free part for 20 min to initiate cell division. After additional incubation, the arrest of lymphocytes at metaphase in both cultures was initiated by incubation with colcemid for 1 h. Peripheral lymphocytes then were harvested, spread, and stained as described in [18]. Two chromosome aberration yields were set for each individual: standard G2 yield (caffeine-free) and next AT imitating high radiosensitivity level yield based on a caffeine-induced G2-M checkpoint arrest. IRS was considered as the ratio of standard G2 yield to imitated high radiosensitivity level in the case of AT in each patient expressed in percentages (1).
(1)$$\mathrm{IRS}=\frac{G2}{G2\mathrm{caffeine}}\times 100\%$$

Patients were categorized based on the IRS values (<30%—radioresistant, 30% ≤ IRS ≤ 50%—normal, 50% ≤ IRS ≤ 70—radiosensitive, >70%—highly radiosensitive) [19].

### 2.7. Statistical Analysis

To describe characteristics of patients, to compare different patient groups, and to determine the impact of various variables on the repeated measurements the statistical analysis was used. Scale variables were described by median and standard deviation. Categorical variables were described by frequency of distribution. Normality assumption was verified with the Shapiro–Wilk test. ANOVA or the Kruskal–Wallis test was used to determine the differences between the means, and the Chi-square test was used to determine the association between variables. In order to determine the impact of different factors on the repeated measurements of ARD, the logistic regression with sorted cases by subject and within-subject variables was used. The sample size for studies with repeated measures was enough to determine the statistical significance of the odds ratio ≥2 with an alpha value equal to 0.05 and power equal to 0.8. Differences were considered statistically significant if a *p*-value was less than 0.05. The statistical analysis was performed using R i386 4.0.5 (R Foundation for Statistical Computing, Vienna, Austria) and IBM SPSSstatistics 21 (Armonk, NY, USA).

## 3. Results

### 3.1. Patients Characteristics

IRS was successfully determined for 91 out of 108 patients who signed the informed consent form. None of these patients were radioresistant, 10 patients (11%) had normal IRS, 47 patients (52%) were categorized as radiosensitive, and 34 (37%) as highly radiosensitive. Histograms of IRS distribution in each group are displayed in Appendix A. There were no significant differences among IRS groups according to age, disease stage, hormone therapy, breast volume, skincare, and RT dose (Table 1).

There were no significant differences among patient groups with different skincare according to age, disease stage, hormone therapy, breast volume, IRS, and RT dose (Table 2).

### 3.2. ARD Manifestation

Clinical manifestation of ARD was evaluated weekly by three radiation oncologists independently according to the CTCAE scale. The presence of moist desquamation (MD) was evaluated during RT and up to 4 weeks after.

All patients had normal skin at the beginning of RT. There were no significant differences in the manifestation of ARD between patient groups with different IRS (Table 3) as well as in different skincare groups (Table 4).

### 3.3. Evaluation of Demographic and Clinical Characteristics as Significant Factors of ARD Assessed by CTCAE Scale

Logistic regression was used to evaluate the impact of different factors on repeated measurements of ARD assessed by the CTCAE scale (Table 5).

In this model, repeated measurements of ARD were included from 10 fx when 40.7% of all patients had IR induced dermatological symptoms. The degree of ARD according to the CTCAE scale from 10 fx up to the end of RT was assigned as a response (CTCAE scale from 10 fx to 25 fx in Table 5). A univariate logistic regression revealed a statistically significant impact of age, breast volume, and disease stage on ARD manifestation. According to the results, older patients with bigger breasts and higher than 0 disease stage were prone to more severe ARD (Table 5).

A multivariate logistic regression showed that only the breast volume had a significant impact on ARD manifestation (Table 6).

## 4. Discussion

After the patient’s agreement to participate in this study, the blood samples for the IRS assessment were collected. For all patients included in this study, we used a modified G2 assay supplemented by a caffeine-induced G2 checkpoint abrogation in order to determine IRS. In our study, 89% of the patients were radiosensitive or highly radiosensitive, 11% of the patients had normal IRS and no patient was radioresistant (Table 1). The proportion of increased radiosensitive patients (89%) in this study is larger compared with numbers reported by other authors [20,21,22]. 

The high number of radiosensitive patients in our study can be associated with the particularity of the used radiosensitivity test. This study employed a modified G2 chromosome sensitivity assay, where the G2 checkpoint efficiency is analyzed instead of the standard G2 aberration yield. The example-based explanation of the differences between conventional and modified G2 assays is provided in the Appendix A (see text in Appendix A) with a comprehensive description of depicted data. The main idea of the suggested adjustment of G2 assay is to mimic Ataxia telangiectasia syndrome and compare the efficiency of ATM reparation within a single patient when this pathway is “on” and caffeine-induced is “off”. Consequently, the obtained results with a high number of radiosensitive individuals can be related to the variation ATM association with higher breast cancer risk. ATM disorders’ involvement in increased breast cancer risk reported in [23].

Another study discusses considerably low risk for breast cancer and clinical radiosensitivity but suggests that there is some threshold for the ATM protein leading to a high risk of radiosensitivity for breast cancer patients [24]. There are a lot of mixed and even contradicting results in the field of investigation of factors responsible for radiosensitivity including ATM. K.J. Jerzak et al., in detail discussed different data about radiation therapy outcome in the case of different types of ATM pathology: it was found that some data suggest an increase in toxicity, other data suggest, radiation-induced clinical benefit; some studies propose that ATM mutations might increase the risk of breast cancer after RT, others say that ATM status should not be used to make treatment decisions with respect to radiotherapy [25]. Therefore, more extensive studies in this regard are required and we are in the scope of this way.

Before the IRS assessment, patients were randomized into two groups according to skincare. The best supportive skincare was prescribed for all patients during RT. After randomization, additional prophylactic skincare with a plain, perfumeless, moisturizing cream containing 40% water was prescribed twice a day for the second group of patients. The aim of this prescription was to check the hypothesis that additional skin hydration can reduce IR induced acute skin reaction [14]. There were no statistically significant differences in patient characteristics between skincare groups (Table 2). The results of our study showed no significant differences in the manifestation of ARD in these groups as well (Table 4).

The logistic regression showed no influence of additional skincare for the repeated measurements of ARD assessed by the CTCAE scale (Table 5). Whereas the mean IRS in the BSC patient group was 64.6%, and 65.8% in the ASC group (*p* = 0.7), and patients with different IRS categories were distributed equally among skincare groups (Table 2). To summarize, the additional skincare did not improve the skin condition during RT, and IRS was distributed equally among patients with different skincare. Therefore, in a further analysis, we investigated if there were any other factors that can influence the clinical manifestation of ARD.

There were no statistically significant differences among different IRS patient groups comparing various characteristics such as age, disease stage, use of tamoxifen (hormonotherapy), breast size, or RT parameters (Table 1). However, there were some important differences. In our study, patients with normal IRS were older and had bigger breasts than radiosensitive patients.

The mean age of patients in the normal IRS group (60.62 ± 10.54 y) was higher by 2.71 years compared with the mean age in the radiosensitive patient group and higher by 7.11 years compared with the mean age in the highly radiosensitive patient group (Table 1). Some authors claim that age is an important factor for radiosensitivity, and patients between 40 and 50 years tend to be more radiosensitive [26,27]. Our data confirm this statement because the highest mean percentage of IRS (70.4%) was observed in the subgroup of patients from 40 to 50 years old (Table A1 in Appendix B).

The breast volume of patients was calculated from RT plans using the clinical target volume (CTV) structure, which encompasses whole breast tissue. The mean breast volume in the normal IRS patient group (951 ± 481 mL) was bigger by 85 mL compared with the mean breast volume in the radiosensitive patient group, and by 150 mL compared with the mean breast volume in the highly radiosensitive patient group. This difference is important as we know from the literature that a bigger breast, as well as a higher body mass index, are risk factors for ARD [28,29,30].

There were no significant differences in the manifestation of ARD between patient groups with different IRS (Table 3). The absence of a clear correlation between IRS and ARD can also be addressed to the phenomenon such as adaptive response and in the contest of radiation effects called “hormesis”. Previously it was found that human lymphocytes reattaining their viability can modify reparation rate in response to low-dose ionizing radiation initiating either adaptation or hypersensitivity as well as changes in sensitivity to ionizing radiation in human tumor cell lines expressed as hypersensitivity at low doses followed by induced radioresistance at higher doses [31]. Further, this phenomenon was even proposed to be employed for the improvement of radiotherapy [32]. Moreover, it was demonstrated by the RIANS models described in [33,34].

ATM cascades play an important role in the initiation of hormesis which can affect organism response to further radiation. This means that IRS may not remain constant during the treatment due to repeatable exposure to low doses evoking an adaptive response or hypersensitization to further irradiation thus modifying clinical outcome. Therefore, it would be feasible to apply the multi testing approach during the treatment to investigate the pattern of IRS alteration after a different number of pre-exposure and at the end of the treatment to reveal along with research on mechanisms underlying individual susceptibility to initiate adaption or sensitization of an organism to ionizing radiation in response to low dose. A pilot study with prostate cancer patients observed the alteration in IRS during the treatment with the reflection in differences in clinical outcome depending on it [35].

A univariate logistic regression showed that age, breast volume, and disease stage had a statistically significant influence on the repeated measurements of ARD (Table 5).

Multivariate logistic regression showed that breast volume had the most significant negative influence on the manifestation of ARD (Table 6). These results call for the consideration of other possible radiotherapy methods for carefully selected patients. For example, patients with big breasts can be treated with the hypofractionation technique or using intraoperative RT. Hypofractionated radiotherapy for early-stage breast cancer is associated with a decrease in acute skin toxicity [36] and does not reduce disease control [37]. Intraoperative partial breast irradiation significantly reduces the dose for healthy tissues [38] and is a suitable treatment option for low-risk patients with a low local recurrence rate [39].

## 5. Conclusions

The additional skincare did not improve skin condition during RT. A total of 89% of patients had increased radiosensitivity. IRS determined before RT did not show the predictive value for the manifestation of ARD. Logistic regression revealed that breast volume was the most significant risk factor for the manifestation of ARD.

## Figures and Tables

**Table 1 life-12-00020-t001:** Patient’s characteristics in different IRS groups.

Characteristics		Normal IRS (30% ≤ IRS ≤ 50%)	Radiosensitive (>50% ≤ IRS ≤70%)	Highly Radiosensitive (IRS > 70%)	*p*
Patients	N (%)	10 (11%)	47 (52%)	34 (37%)	
Age (y)	Mean, SD	60.62 ± 10.54	58.01 ± 8.48	55.08 ± 8.88	0.18
	median	59.85	57.5	55.2	
	Min–max	39.6–75.5	31.0–75.7	28.3–71.5	
Stage	0	3 (3.3%)	15 (16.5%)	5 (5.5%)	0.08
	I	3 (3.3%)	22 (24.2%)	22 (24.2%)	
	II	4 (4.4%)	10 (11.0%)	7 (7.7%)	
HT	No	2 (2.2%)	16 (17.6%)	9 (9.9%)	0.68
	Yes	8 (8.8%)	31 (34.1%)	25 (27.5%)	
BV	Mean, SD	951 ± 481	866 ± 477	800 ± 379	0.68
Skincare	BSC	5 (5.5%)	26 (28.6%)	13 (14.3%)	0.31
	ASC	5 (5.5%)	21 (23.1%)	21 (23.1%)	
Dose (Gy)	CTV Dmean, SD	48.93 ± 0.96	48.37 ± 1.24	49.11 ± 1.64	0.20
	HI	0.43 ± 0.13	0.46 ± 0.10	0.45 ± 0.12	0.74
	CTV Dmax, SD	54.1 ± 0.15	53.8 ± 0.98	54.6 ± 1.45	0.29

IRS—individual radiosensitivity, y—years, N—number, SD—standard deviation, Min—minimum, max—maximum, HT—hormonotherapy, BV—breast volume, ml—milliliters, BSC—best supportive care, ASC—additional skincare, Gy—grays, Dmean—mean dose, HI—homogeneity index, Dmax—maximum dose.

**Table 2 life-12-00020-t002:** Patient’s characteristics in different skincare groups.

Characteristics		BSC	ASC	*p*
Patients	N (%)	44 (48.8%)	47 (51.6%)	
Age (y)	Mean, SD	58.6 ± 7.5	56.0 ± 10.1	0.17
	median	59.0	56.0	
	Min–max	40.6–75.7	28.3–75.2	
Stage	0	9 (9.9%)	14 (15.4%)	0.08
	I	28 (30.8%)	19 (20.9%)	
	II	7 (7.7%)	14 (15.4%)	
HT	No	12 (13.2%)	15 (16.5%)	0.63
	Yes	32 (35.2%)	32 (35.2%)	
BV	Mean, SD	864 ± 441	837 ± 444	0.99
IRS	Mean, SD	64.6 ± 11.9	65.8 ± 10.6	0.70
Dose (Gy)	CTV Dmean, SD	48.6 ± 1.55	48.8 ± 1.28	0.51
	HI	0.44 ± 0.11	0.46 ± 0.12	0.61
	CTV Dmax, SD	54.1 ± 1.29	54.2 ± 1.11	0.85

BSC—best supportive care, ASC—additional skincare, y—years, N—number, SD—standard deviation, Min—minimum, max—maximum, HT—hormonotherapy, BV—breast volume, IRS—individual radiosensitivity, ml—milliliters, Gy—grays, Dmean—mean dose, HI—homogeneity index, Dmax—maximum dose.

**Table 3 life-12-00020-t003:** ARD manifestation in different IRS groups.

Characteristics		Normal IRS (30% ≤ IRS ≤ 50%)	Radiosensitive (>50% ≤ IRS ≤70%)	Highly Radiosensitive (IRS > 70%)	*p*
CTCAE 5 fx	0	9 (9.9%)	40 (44.0%)	30 (33.0%)	0.91
	I	1 (1.1%)	7 (7.7%)	4 (4.4%)	
CTCAE 10 fx	0	3 (3.3%)	28 (30.8%)	24 (26.4%)	0.15
	I	7 (7.7%)	19 (20.9%)	9 (9.9%)	
	II	0 (0%)	0 (0%)	1 (1.1%)	
CTCAE 15 fx	0	0 (0%)	9 (9.9%)	9 (9.9%)	0.38
	I	10 (11.0%)	37 (40.7%)	24 (26.4%)	
	II	0 (0%)	1 (1.1%)	1 (1.1%)	
CTCAE 20 fx	0	0 (0%)	2 (2.2%)	3 (3.3%)	0.91
	I	6 (6.6%)	38 (41.8%)	21 (23.1%)	
	II	4 (4.4%)	7 (7.7%)	10 (11.0%)	
CTCAE 25 fx	I	3 (3.3%)	22 (24.2%)	16 (17.6%)	0.88
	II	7 (7.7%)	24 (26.4%)	17 (18.7%)	
	III	0 (0%)	1 (1.1%)	1 (1.1%)	
MD	No	9 (9.9%)	37 (40.7%)	26 (28.6%)	0.81
	Yes	1 (1.1%)	10 (11.0%)	8 (8.8%)	

CTCAE—common terminology criteria for adverse events, fx—fractions, IRS—individual radiosensitivity, MD—moist desquamation.

**Table 4 life-12-00020-t004:** ARD manifestation in different skincare groups.

Characteristics		BSC	ASC	*p*
CTCAE 5 fx	0	37 (40.7%)	42 (46.2%)	0.33
	I	7 (7.7%)	5 (5.5%)	
CTCAE 10 fx	0	27 (29.7%)	28 (30.8%)	0.56
	I	16 (17.6%)	19 (20.9%)	
	II	1 (1.1%)	0 (0%)	
CTCAE 15 fx	0	9 (9.9%)	9 (9.9%)	0.32
	I	33 (36.3%)	38 (41.8%)	
	II	2 (2.2%)	0 (0%)	
CTCAE 20 fx	0	4 (4.4%)	1 (1.1%)	0.34
	I	30 (33.0%)	35 (38.5%)	
	II	10 (11.0%)	11 (12.1%)	
CTCAE 25 fx	I	21 (23.1%)	20 (22.0%)	0.37
	II	23 (25.3%)	25 (27.5%)	
	III	0 (0%)	2 (2.2%)	
MD	No	33 (36.3%)	39 (42.9%)	0.25
	Yes	11 (12.1%)	8 (8.8%)	

CTCAE—common terminology criteria for adverse events, fx—fractions, MD—moist desquamation; BSC—best supportive care, ASC—additional skincare.

**Table 5 life-12-00020-t005:** The impact of single variables to the repeated measurements of ARD.

Response	Explanatory Variables	Odds Ratio (95% CI)	*p*
ARD degree according to CTCAE scale from 10 fx to 25 fx	Scale variables		
	Age, y	0.97 (0.95–1.00)	0.046
	Breast volume, ml	0.99 (0.99–0.99)	<0.001
	IRS, %	1.02 (0.99–1.04)	0.14
	Breast Dmean, Gy	0.94 (0.82–1.08)	0.39
	Breast Dmax, Gy	0.93 (0.79–1.09)	0.38
	Breast DHI	2.66 (0.41–17.38)	0.31
	Skin Dmean, Gy	1.07 (0.95–1.21)	0.29
	Skin Dmax, Gy	1.04 (0.89–1.21)	0.61
	Skin DHI	3.58 (0.372–34.47)	0.29
	ER, %	0.99 (0.99–1.00)	0.17
	PR, %	0.99 (0.99–1.00)	0.48
	Categorical variables		
	Stage	2.09 (1.024–4.261)	0.04
	Additional skincare	0.93 (0.59–1.46)	0.74
	HT	0.85 (0.53–1.37)	0.52

CTCAE—common terminology criteria for adverse events, IRS—individual radiosensitivity, Dmean—mean dose, Dmax—maximal dose, DHI—dose homogeneity index, ER—estrogen receptors, PR—progesterone receptors, HT—hormonotherapy.

**Table 6 life-12-00020-t006:** Multivariate regression indicating that only breast volume had significant impact to repeated measurements of radiation dermatitis.

Response	Explanatory Variables	Odds Ratio (95% CI)	*p*
ARD degree according to CTCAE scale from 10 fx to 25 fx	Scale variables		
	Age, y	1.01 (0.98–1.04)	0.64
	Breast volume, ml	0.99 (0.99–1.00)	0.002
	IRS, %	1.02 (0.99–1.04)	0.18
	Breast Dmean, Gy	1.28 (0.79–2.05)	0.31
	Breast Dmax, Gy	0.67 (0.44–1.02)	0.06
	Breast DHI	4.22 (0.07–238.9)	0.49
	Skin Dmean, Gy	1.12 (0.83–1.50)	0.46
	Skin Dmax, Gy	0.97 (0.74–1.28)	0.84
	Skin DHI	10.52 (0.07–1591.52)	0.36
	ER, %	1.01 (0.9913–1.0113)	0.62
	PR, %	1.01 (0.9915–1.01)	0.78
	Categorical variables		
	Stage >0	0.55 (0.29–1.04)	0.07
	HT	1.44 (0.88–2.36)	0.15
	Additional skincare	0.80 (0.51–1.24)	0.80

CTCAE—common terminology criteria for adverse events, IRS—individual radiosensitivity, Dmean—mean dose, Dmax—maximal dose, DHI—dose homogeneity index, ER—estrogen receptors, PR—progesterone receptors.

## Data Availability

All data will be made available upon reasonable request to the corresponding author.

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
