# Peer review of "Individual Radiosensitivity as a Risk Factor for the Radiation-Induced Acute Radiodermatitis"

_life, 2021, doi:10.3390/life12010020_

Round 1

Reviewer 1 Report

This work is serious in most methodological approaches. The number of patients studied and clinical manifestation of ARD evaluated weekly by three radiation oncologists, are particularly valuable.

However, the weak point of this study seems to me to be the most crucial: the constitution of the radiosensitivity groups and lead to wrong interpretation of the results.

1) the percentage of radiosensitive or even hyperradiosensitive cases (89%) seems extremely high.  The justification given in one line by the authors is too weak. Their argument concerning this crucial point should be discussed more extensively and supported by several other studies.

2) the method of evaluation of IRS statut has to be described most extensively and consolidate by published references. The used test (G2 arrest test) is not an appropriated test for radioresistance but is most a test for radio-susceptibility (PMID: 34281212).

3) I suggest that the authors cross-reference their test with other types of tests to define radiosensitivity (e.g., the gold standard : Clonogenic assay or alternative test of ATM PMID: 34069662, PMID: 32425719) in order to verify their starting group and to revise or confirm their conclusion on the absence of influence of IRS status on the occurrence of ARD.

Reviewer 2 Report

Calculate appropriate sample size and adjust the other cofounders. Conduct the study on larger sample size. Cite more supportive references.

Reviewer 3 Report

This study aimed to evaluate the clinical manifestation of acute radiation dermatitis (ARD) in different skincare groups during whole breast radiotherapy depending on IRS and other risk factors. This study was well designed and presented. I only have some comments to the authors.

1. I suggest the authors to further analyze the expression level of 15 radioresponsive genes (ATM, BAX, BBC3, BCL2, CCNG1, cMYC, DDB2, FDXR, GADD45A, MDM2, CDKN1A, PCNA, SESN1, XPC, and ZMAT3 genes) and evaluate their associations with skin reactions.

2. Patients with larger breast size may also have greater skin fold within the radiation field, and the radiation dermatitis are often more severe at these skin folds. Could the authors describe and explain the location of radiation dermatitis?

3. The definition of breast volume should be described more clearly in the methods. The authors should also explain why hypofractionated breast radiotherapy were not delivered.

Round 2

Reviewer 1 Report

Thank you for the answers and the additions in the manuscript that bring clarity. 
The method and comparative table used in the answer (or equivalent) deserves to appear in the data supplement.
Apart from this point, the rest seems okay.

Author Response

Dear reviewer, according to your comment we added a more detailed method description with a comparative table in supplementary data (see attached file). The Discussion was also supplemented with this text: "The example-based explanation of the differences between conventional and modified G2 assays is provided in supplementary data (see text in supplementary data and table 2) with a comprehensive description of depicted data".

Reviewer 2 Report

I do not see the methodology section edited to match the authors' response sheet. Sample size justification should be added under the statistical section.

See the attached document for more details.

Author Response

Dear reviewer, according to your comment this information was added to the Statistical analysis subsection: "The sample size for studies with repeated measures was enough to determine the statistical significance of odds ratio ≥ 2 with an alpha value equal to 0.05 and power equal to 0.8".

Reviewer 3 Report

The authors had addressed my concerns.

Author Response

Dear reviewer, thank you so much for your time reading and analyzing our work. We truly believe that your comments helped to improve the manuscript a lot.